# Curve-Fitting Correction Method for the Nonlinear Dimming Response of Tunable SSL Devices

**Rugved Kore * and Dorukalp Durmus**

Department of Architectural Engineering, Pennsylvania State University, University Park, PA 16802, USA; alp@psu.edu
* Correspondence: rsk97@psu.edu

**Abstract:** Solid-state lighting (SSL) devices are ubiquitous in several markets, including architectural, automotive, healthcare, heritage conservation, and entertainment lighting. Fine control of the LED light output is crucial for applications where spectral precision is required, but dimming LEDs can cause a nonlinear response in its output, shifting the chromaticity. The nonlinear response of a multi-color LEDs can be corrected by curve-fitting the measured data to input dimming controls. In this study, the spectral output of an RGB LED projector was corrected using polynomial curve fitting. The accuracy of four different measurement methods was compared in order to find the optimal correction approach in terms of the time and effort needed to perform measurements. The results suggest that the curve fitting of very high-resolution dimming steps ($n$ = 125) significantly decreased the chromaticity shifts between measured (actual) and corrected spectra. The effect size between approaches indicates that the curve-fitting of the high-resolution approach ($n$ = 23) performs equally well as at very high resolution ($n$ = 125). The curve-fitting correction can be used as an alternative approach or in addition to existing methods, such as the closed-loop correction. The curve fitting method can be applied to any tunable multi-color LED lighting system to correct the nonlinear dimming response.

**Keywords:** solid-state lighting; light output control; smart lighting system; building technologies

## 1. Introduction

Solid-state lighting (SSL) devices are ubiquitous in architectural, industrial, and commercial lighting applications due to their compactness, durability, long life expectancy, and spectral tunability. Although phosphor-converted LEDs (pcLEDs) are currently the most prevalent SSL devices, multi-primary (or multi-channel, multi-color, color-mixed) LEDs, OLEDs, and quantum dots offer unparalleled flexibility in terms of spectral power distribution (SPD) [1–4]. The multi-color SSL device SPDs can be optimized for energy efficiency, color rendition, art conservation, circadian entrainment, cyanosis detection, tissue visualization in surgery, plant growth, and ecological conservation [5–12].

In addition to architectural luminaires and display technologies, multi-color (e.g., red, green, and blue (RBG)) LED projectors are used for commercial applications, such as reducing damage to artwork while maintaining the perceived color quality [7,8]. However, all the past studies optimizing light source SPDs assume that light source outputs are linear (e.g., dimming an LED channel 50% provides 50% power output of that channel). In reality, LEDs may exhibit a nonlinear response when dimmed. LEDs' nonlinear response can be a result of aging, warm-up, or variations in junction temperature. Various reasons, such as package architecture, materials, environment, thermal management, and control system can cause considerable variations in junction temperature [13–20]. The temperature variation affects the diode output and the variation in intensity and peak wavelength during warmup and dimming [21]. The nonlinear response inhibits the fine control of the spectra, which is crucial for applications where fine spectral control is required, such as circadian entrainment or cyanosis detection in hospitals and health care centers, color quality in

museums, art galleries, and retail spaces, imaging for medical and art conservation, and horticultural applications.

Thermal junction variation is one of the primary reasons for nonlinearity, and the negative effects can be reduced using loop systems with thermal feedback. For example, Llenas et al. [22] used a proportional–integral–derivative controller system (PID) to fine-tune pulse-width modulation (PWM) weighting factors to compensate for small spectral changes in the LEDs caused by junction thermal variation or nonlinear response. The diode junction temperature can be managed by the loop systems via thermal feedback to address the chromaticity shifts [17]. Several types of loop systems have been developed to address this requirement, such as open-loop (OL) temperature feed-forward (TFF), flux feedback (FFB), color coordinates feedback (CCFB), and FBB&TFF [21–27]. Many LED systems are being manufactured with integrated microcontrollers to handle various dimming curves [18]. However, these solutions can be complex and increase the cost of luminaires, which may cause additional failure modes [27]. Effective thermal management is among the least expensive approaches to improving color stability, but it needs active cooling to make it most efficient, which adds extra cost, introduces additional failure modes, and limits the use of many luminaires [28]. Also, incorporating these systems in commercially available LEDs without any provision can be challenging.

The nonlinear dimming response of LEDs can also be corrected by curve-fitting the measurement data to dimming levels or using them as another layer of correction with loop systems and integrated microcontrollers. For example, Stefan et al. used the Bussgang theorem to reduce spectral power to a single value (signal-to-noise ratio (SNR)) and performed sixth-order polynomial curve fitting using the least squares curve-fitting technique in a Monte Carlo simulation to address the nonlinear dimming behavior of LEDs while investigating illuminance distribution of white LEDs inside a room under different dimming levels [25]. To control an LED system efficiently in a rail transportation application, Dapeng et al. used curve fitting on illuminance–current and illuminance–voltage data of colored LEDs and obtained a weighting function to modulate the PWM duty ratio [29].

A modulation technique called color-shift keying (CSK) plays a critical role in modern wireless communication. Halder and Barman characterized the dimming nonlinearity of RGB LEDs and performed polynomial curve fitting of the 5th degree to improve symbol error rate performance of the received CSK constellation of chromaticity points, and the field of view (FOV) angle of the LED chip was also increased [30]. To improve the spectral sensitivity of the cameras in an imaging system, Durmus proposed the use of a genetic algorithm to optimize the spectral properties of three theoretical sensors to minimize the error between estimated and actual light spectra [31].

The nonlinear response of two-channel (bicolor) white LEDs (i.e., warm and cool white LED mixture) is well studied. For example, Dyble et al. [26] studied the chromaticity shift in mixed-color LED and pcLED systems due to continuous current system and PWM dimming. Under both dimming systems, white pcLEDs exhibited small chromaticity shifts (<4 MacAdam ellipses) with PWM performing slightly better, whereas mixed-color white LEDs suffered large chromaticity shifts (>4 MacAdam ellipses). The study concluded that if a mixed-color system is required, then an active feedback system must be incorporated to avoid perceivable chromaticity shifts. Past studies aimed to address the nonlinear dimming response of white LEDs (especially spectral shifts due to temperature changes), but they have not investigated multi-color chromatic LEDs. While the changes in correlated color temperature (CCT) and luminous flux are important at a higher level, these metrics have inherent limitations [32]. Therefore, photometric and colorimetric outputs should be accompanied by radiometric measures when analyzing data.

Another critical area for improvement is dealing with a "black box" (an integrated tunable LED that has dedicated active thermal management) that is underperforming and in need of improvements. The aim of the study was to correct the nonlinear dimming response of multi-color LED systems without intervening with the internal mechanisms of the "black box". Such an approach will enable correcting LED dimming shifts in situ.

Therefore, the proposed method aims to correct output independent of the spectral or heating characteristics of the device. Here, the spectral peak output of three primary channels of an RGB projector (LF2+ by Lightform, San Francisco, CA, USA) [33] was measured and corrected using polynomial curve fitting as proof of concept. Since an RGB lighting system can produce around 16 million ($255^3$) combinations, the accuracy of four different correction approaches was compared to find the optimal correction approach in terms of time and effort needed to perform measurements.

## 2. Materials and Methods

The output of each channel of an RGB LED projector (Lightform LF2+) was measured at each dimming step from 0 (LED turned off) to 255 (LED at full output) using a calibrated illuminance spectrophotometer (CL-500A by Konica Minolta, NJ, USA) in the Penn State Lighting Laboratory, as shown in Figure 1. To avoid spectral shifts in the output due to temperature, the measurements were performed 15 min after starting the device. Three separate sets of measurements for each channel were performed and each set required approximately 22 min to complete. The positions of the projector and the spectrophotometer were maintained with zero ambient light as a control variable throughout the experiment. Figure 2 shows the nonlinear responses of the light output of the red LED (measured spectra) at 20%, 40%, 60%, and 80% dimming levels compared to ideal (theoretical) values.

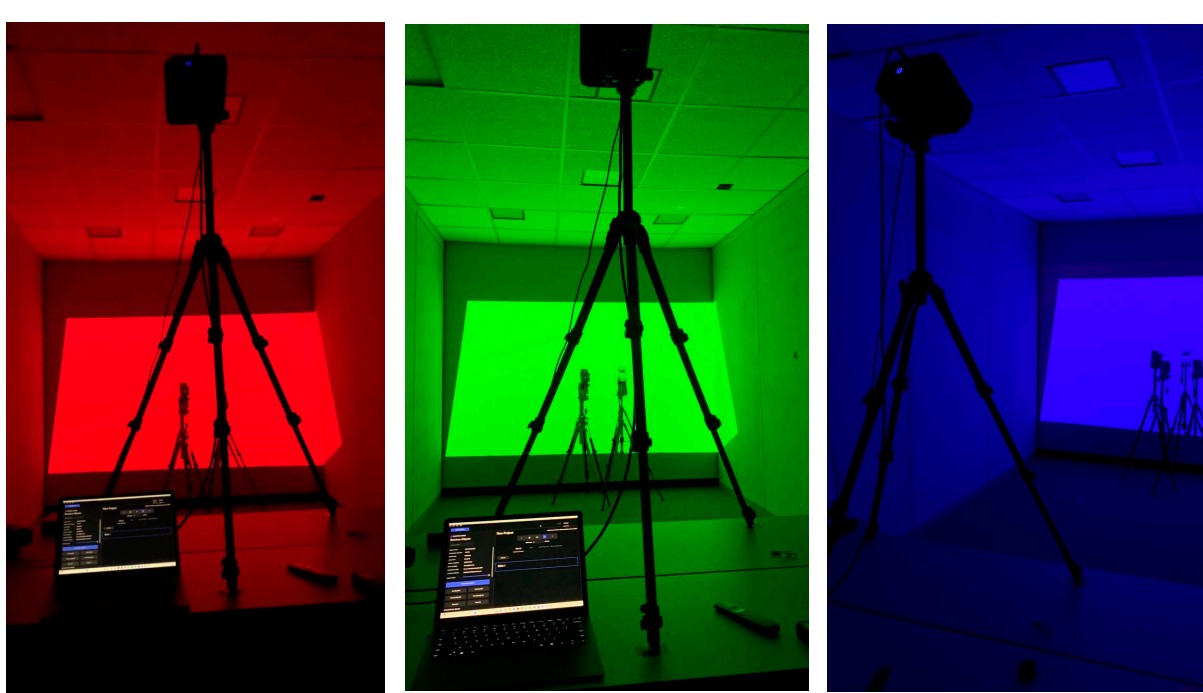

**Figure 1.** Experimental setup used to measure the dimming steps of individual channels of an RGB projector (Lightform LF2+).

First, two curve-fitting parameters were investigated: the choice of scaling reference (50% or 100% dimming level of the measured SPD) and the method for reducing the SPD to a single value (maximum or total of the SPD). The results indicate that using the maximum, or total power to reduce the SPD to a single value, or the choice of reference (100% vs. 50% of the measured SPD), does not affect the cubic dimming curve functions, as shown in Table 1. Here, the spectral output of each channel was reduced to a single value (the maximum radiant power for each SPD) to correlate light output with a dimming step.

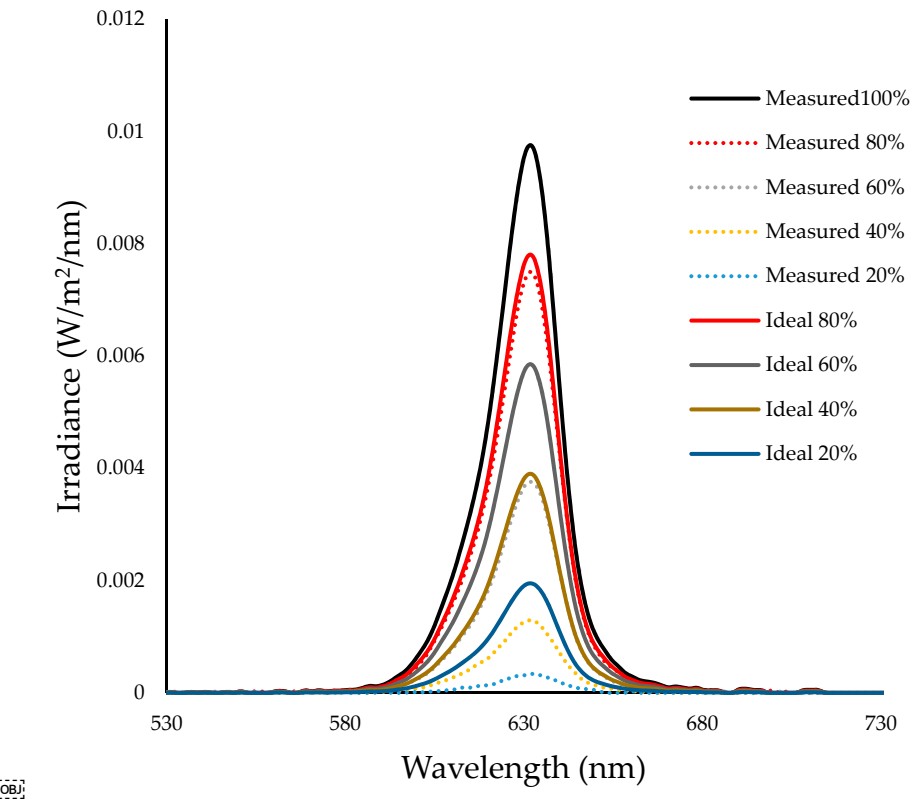

**Figure 2.** The difference between measured and calculated spectral power distribution of the projector's red channel at four control input levels (20%, 40%, 60%, and 80% of the total light output).

**Table 1.** Curve-fitting formulae and correlation coefficients to analyze the impacts of the SPD reduction technique.

| Light Output | Comparison | Maximum Power | Total Power |
| --- | --- | --- | --- |
| **100%** | Estimated vs. measured SPDs | $y = 37.41\,x^3 - 13.108\,x^2 + 1.8273\,x + 0.011$ <br> $R^2 = 0.9675$ | $y = 0.0509\,x^3 - 0.5062\,x^2 + 1.9665\,x$ <br> $+ 0.1425$ <br> $R^2 = 0.9669$ |
| | Dimming level vs. measured SPDs | $y = -6 \times 10^{-8}\,x^3 + 2 \times 10^{-5}\,x^2$ <br> $- 0.0011\,x + 0.0165$ <br> $R^2 = 0.9959$ | $y = -2 \times 10^{-6}\,x^3 + 0.0007\,x^2 - 0.0303\,x$ <br> $+ 0.5577$ <br> $R^2 = 0.9959$ |
| **50%** | Estimated vs. measured SPDs | $y = 43.068\,x^3 - 15.091\,x^2 + 2.1037\,x$ <br> $+ 0.0126$ <br> $R^2 = 0.9675$ | $y = 0.0593\,x^3 - 0.5899\,x^2 + 2.2916\,x$ <br> $+ 0.1661$ <br> $R^2 = 0.9669$ |
| | Dimming level vs. measured SPDs | $y = -6 \times 10^{-8}\,x^3 + 2 \times 10^{-5}\,x^2$ <br> $- 0.0011\,x + 0.0165$ <br> $R^2 = 0.9959$ | $y = -2 \times 10^{-6}\,x^3 + 0.0007\,x^2 - 0.0303\,x$ <br> $+ 0.5577$ <br> $R^2 = 0.9959$ |

However, spectral measurements at lower dimming steps from 0 to 56 were removed due to high noise levels in this dimming range and their inability to produce enough light for photopic vision. The spectral measurements were analyzed to simulate four different measurement correction approaches (i.e., resolutions):

(1) Very high resolution: using measurement points from every alternate dimming step (e.g., 255, 253, 251, 249 . . .) resulting in $n = 125$ measurement points per LED channel;
(2) High resolution: using measurement points from every 10th dimming step (e.g., 255, 245, 235, 225 . . .) resulting in $n = 23$ measurement points per LED channel;
(3) Medium resolution: using measurement points from every 20th dimming step (e.g., 255, 235, 215, 195 . . .) resulting in $n = 12$ measurement points per LED channel;

(4)  Low resolution: using measurement points from every 50th dimming step (e.g., 255, 205, 155, 105 . . .) resulting in *n* = 6 measurement points per LED channel.

Third-order polynomial curve fitting was performed to generate equations for each of the four measurement resolution approaches. Figure 3 shows the third-order polynomial curve fitting for red channel peak power for a very high-resolution correction approach (alternate dimming steps) with *n* = 125 measurement points per channel. The equations were then used to generate corrected SPDs at each step from 56 to 255. Consequently, the chromaticity differences between the corrected SPDs under four resolutions were compared to actual (measured) SPDs at each dimming step using $\Delta u'v'$ in the CIE 1976 ($u'$ $v'$) chromaticity diagram [34] and root mean square error (RMSE). In addition to corrected SPDs, the theoretical (linear) response of LEDs was also calculated and compared against the measured (actual) SPDs. All the calculations were performed in Matlab$^{\circledR}$.

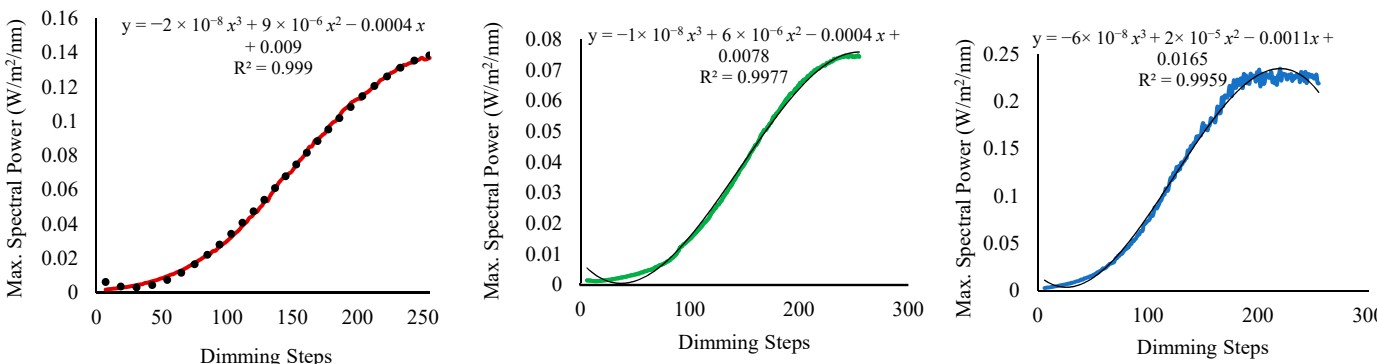

**Figure 3.** A third-order polynomial very high-resolution curve fitting (black dotted line) for the nonlinear dimming response of a red LED channel peak power (red continuous line). The solid black line shows the ideal (linear) dimming response.

It is reasonable to expect that a very high-resolution correction approach would require more time due to the number of measurements but will result in higher colorimetric, photometric, and radiometric accuracy in accounting for response nonlinearity. On the other hand, a low-resolution approach would save users time since only six measurements per channel are needed to correct the nonlinear output of an LED system. It is reasonable to hypothesize that four correction approaches will perform better than theoretical (linear) dimming response, but there will be a plateau in accuracy for correction methods (e.g., increasing the number of measurement points will not increase the photometric accuracy while requiring more time and effort to collect data).

## 3. Results

The chromaticity differences ($\Delta u'v'$) between the corrected and measured SPDs, and theoretical (linear) and measured SPDs, are shown in Figure 4. The difference between measured and theoretical SPDs was the largest whereas the chromaticity difference was the smallest for the very high-resolution correction approach (*n* = 125). The chromaticity difference increased with a decrease in dimming steps for curve fitting (*n* = 23, *n* = 12, and *n* = 6).

A one-sample Kolmogorov–Smirnov (KS) test was performed to check the normality of the data, and the data were not normally distributed. Since non-parametric tests do not assume that data are approximately normally distributed and they are based on fewer assumptions, the Wilcoxon rank-sum test was used to find the statistical significance in chromaticity difference. Table 2 shows that there was a significant difference between each correction approach and the corrected SPDs and theoretical SPDs ($p < 0.001$). The results indicate that the alternate step correction (*n* = 125) provides the highest chromaticity accuracy. The effect size between very high resolution (*n* = 125) and high resolution (*n* = 23) was small (*r* = 0.3), which indicates that high resolution was a good alternative to a very-

high-resolution approach. The effect size between very high resolution ($n$ = 125) and medium resolution ($n$ = 12) was medium ($r$ = 0.41), and the effect size between very high resolution ($n$ = 125) and low resolution ($n$ = 6) was large ($r$ = 1.03), as shown in Table 3.

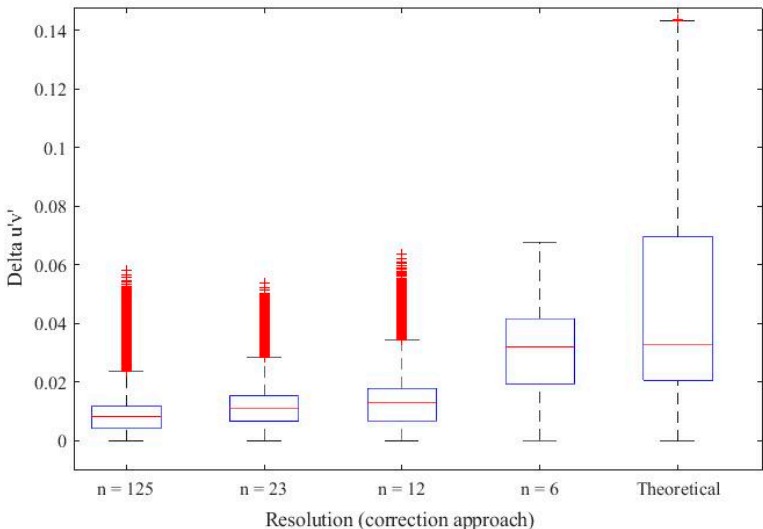

**Figure 4.** Chromaticity difference ($\Delta u'v'$) between four correction approaches with different resolutions and measured SPDs.

**Table 2.** The $p$-value for $\Delta u'v'$ between correction approaches and theoretical (linear) dimming.

| $p$-Value | Very High Resolution ($n$ = 125) | High Resolution ($n$ = 23) | Medium Resolution ($n$ = 12) | Low Resolution ($n$ = 6) | Theoretical (Linear) |
|---|---|---|---|---|---|
| Very high resolution ($n$ = 125) | - | | | | |
| High resolution ($n$ = 23) | <0.00001 | - | | | |
| Medium resolution ($n$ = 12) | <0.001 | <0.00001 | - | | |
| Low resolution ($n$ = 6) | <0.001 | <0.001 | <0.001 | - | |
| Theoretical (linear) | <0.001 | <0.001 | <0.001 | <0.00001 | - |

**Table 3.** The effect size ($d$) for $\Delta u'v'$ between correction approaches and theoretical (linear) dimming.

| Effect Size ($r$) | Very High Resolution ($n$ = 125) | High Resolution ($n$ = 23) | Medium Resolution ($n$ = 12) | Low Resolution ($n$ = 6) | Theoretical (Linear) |
|---|---|---|---|---|---|
| Very high resolution ($n$ = 125) | - | | | | |
| High resolution ($n$ = 23) | 0.3 | - | | | |
| Medium resolution ($n$ = 12) | 0.41 | 0.13 | - | | |
| Low resolution ($n$ = 6) | 1.03 | 0.91 | 0.83 | - | |
| Theoretical (linear) | 1.04 | 0.95 | 0.89 | 0.2 | - |

Although the CIE 1976 $u'v'$ chromaticity coordinates are representative of the visual appearance of a light source, the color spaces used for chromaticity do not take luminance information into account. Since $u'v'$ chromaticity coordinates provide only relative colorimetric information (not photometric or radiometric), they do not provide a comprehensive understanding of the accuracy of LED dimming. Therefore, it is important to consider both metrics for a better comparison.

The RMSE result data were not normally distributed, and the Wilcoxon rank-sum test was used to find statistical significance. The RMSEs between all the correction approaches were significantly different, except for low resolution ($n = 6$) and theoretical SPDs, as shown in Table 4. The RMSE result aligns with $u'v'$ chromaticity coordinates such that the very high-resolution correction ($n = 125$) provides the highest chromaticity accuracy and the effect size between very high-resolution correction ($n = 125$) and high-resolution correction ($n = 23$) was small ($r = 0.23$), which indicates that high-resolution correction was a good alternative to higher resolution correction approaches.

**Table 4.** The *p*-value for RMSE between correction approaches and theoretical (linear) dimming.

| *p*-Value | Very High Resolution ($n = 125$) | High Resolution ($n = 23$) | Medium Resolution ($n = 12$) | Low Resolution ($n = 6$) | Theoretical (Linear) |
|---|---|---|---|---|---|
| Very high resolution ($n = 125$) | - | | | | |
| High resolution ($n = 23$) | <0.00001 | - | | | |
| Medium resolution ($n = 12$) | <0.001 | <0.00001 | - | | |
| Low resolution ($n = 6$) | <0.001 | <0.001 | <0.001 | - | |
| Theoretical (linear) | <0.001 | <0.001 | <0.001 | 0.44 | - |

Figure 4 displays the $\Delta u'v'$ values representing the chromaticity difference between the corrected SPDs under four different resolution approaches and the measured SPDs. The results reveal that the chromaticity difference was smallest for the very high-resolution correction approach ($n = 125$), while it increased with a decrease in dimming steps for the curve-fitting methods with lower resolutions ($n = 23$, $n = 12$, $n = 6$). This indicates that the very high-resolution approach provides the highest chromaticity accuracy, even as dimming levels decrease.

Table 2 presents the *p*-values obtained from Wilcoxon rank-sum tests to assess the statistical significance of the chromaticity differences between the correction approaches and theoretical (linear) dimming. The results demonstrate a significant difference between each correction approach and both the corrected SPDs and theoretical SPDs ($p < 0.001$). These findings affirm that the alternate step correction with very high resolution ($n = 125$) delivers the highest chromaticity accuracy.

Furthermore, Table 3 displays the effect sizes ($r$) for $\Delta u'v'$ between the correction approaches and theoretical (linear) dimming. The effect size between very high resolution ($n = 125$) and high resolution ($n = 23$) was relatively small ($r = 0.3$), indicating that high resolution provides a reasonable alternative to the very high-resolution approach. In contrast, the effect size between very high-resolution correction ($n = 125$) and medium resolution ($n = 6$) and the effect size between very high-resolution correction ($n = 125$) and low resolution ($n = 6$) were substantial ($r = 1.03$ and $1.04$, respectively). These results suggest that diminishing returns may be observed in terms of photometric accuracy with increasing measurement points.

It is essential to note that while the CIE 1976 $u'v'$ chromaticity coordinates provide insight into the visual appearance of a light source, this does not incorporate luminance information. The CIE 1976 $u'v'$ chromaticity coordinates exclusively offer relative colorimetric information and do not provide a comprehensive assessment of the accuracy of spectral matches. Therefore, it is imperative to consider both chromaticity and radiometric values for a holistic comparison.

In line with the $\Delta u'v'$ findings, the RMSE analysis provides complementary insights into the accuracy of SPD matches. Table 4 presents the *p*-values from Wilcoxon rank-sum tests for RMSE between the correction approaches and theoretical (linear) dimming. Notably, the RMSEs between all correction approaches were significantly different, except for low resolution ($n = 6$) and theoretical SPDs. This aligns with the $u'v'$ chromaticity

coordinates' results, reaffirming that very high-resolution correction ($n$ = 125) yields the highest chromaticity accuracy.

Table 5 presents the effect sizes ($r$) for RMSE between the correction approaches and theoretical (linear) dimming. These effect sizes indicate that, similar to the $u'v'$ chromaticity coordinates, the high-resolution correction approach ($n$ = 23) serves as a viable alternative to very high resolution ($n$ = 125), with a relatively small effect size ($r$ = 0.23). However, the effect size between very high-resolution correction ($n$ = 125) and medium resolution ($n$ = 12) and the effect size between very high-resolution correction ($n$ = 125) and low resolution ($n$ = 6) were substantial ($r$ = 0.5 and 1.4, respectively). These results emphasize that the very high-resolution approach may offer diminishing returns in terms of photometric accuracy, particularly as the number of measurement points increases.

**Table 5.** The effect size ($d$) for RMSE between correction approaches and theoretical (linear) dimming.

| Effect Size ($r$) | Very High Resolution ($n$ = 125) | High Resolution ($n$ = 23) | Medium Resolution ($n$ = 12) | Low Resolution ($n$ = 6) | Theoretical (Linear) |
|---|---|---|---|---|---|
| Very high resolution ($n$ = 125) | - | | | | |
| High resolution ($n$ = 23) | 0.23 | - | | | |
| Medium resolution ($n$ = 12) | 0.5 | 0.24 | - | | |
| Low resolution ($n$ = 6) | 1.4 | 1.42 | 1.42 | - | |
| Theoretical (linear) | 0.2 | 1.38 | 1.36 | 0.007 | - |

In summary, both the $\Delta u'v'$ and RMSE analyses consistently demonstrate that the very high-resolution correction approach ($n$ = 125) provides the highest chromaticity and photometric accuracy. While high resolution ($n$ = 23) offers a reasonable alternative, the results suggest that further increasing the number of measurement points may not significantly enhance accuracy and may require additional time and effort. These findings underscore the importance of carefully selecting the resolution approach based on specific application requirements and constraints.

## 4. Discussion

The dimming nonlinearity can be corrected using different methods. Open loops, closed loops, PID, and integrated micro-controllers are some of the well-known methods. Unfortunately, these methods are complex and expensive, and they can add additional failure modes, thus making the solutions even more complex. The curve-fitting method provides an inexpensive and simple but manual solution to address LED nonlinearity. Curve fitting can be used as a stand-alone system or added on top of other correction methods for the light sources used for different applications, such as projection mapping, cyanosis detection, art conservation or restoration, visual enhancement, or circadian entrainment.

This study describes the use of curve fitting for multi-color tunable LED light sources. To obtain optimal chromaticity accuracy within minimum time, four different resolution curve-fitting approaches were compared using two different metrics: CIE 1976 $u'v'$ chromaticity coordinates and RMSE. The accuracies of the four different approaches were significantly different, and the effect size calculation was performed to understand the magnitude of the difference. When compared with very high resolution, the high-resolution approach had the least effect size ($u'v'$: $r$ = 0.3; RMSE: $r$ = 0.23) whereas medium resolution had a medium to large ($u'v'$: $r$ = 0.41; RMSE: $r$ = 0.6) effect size. This result suggests that high-resolution ($n$ = 23) curve fitting can be applied to the dimming data to achieve nonlinearity dimming correction with optimal chromaticity accuracy.

The results presented in this study shed light on the potential of curve fitting as a cost-effective and straightforward method for correcting the nonlinear dimming response of LED light sources. By comparing four different resolution curve-fitting approaches, this

research provides insights into the trade-offs between accuracy and measurement effort, which have significant theoretical and practical implications for a range of applications.

The theoretical implications revolve around the choice between complex and expensive methods like open loops, closed loops, PID controllers, and integrated microcontrollers versus the simplicity and cost-effectiveness of curve fitting. While the former methods have been established for addressing nonlinearity, they introduce complexities and potential failure modes. The findings of this study highlight the practicality of curve fitting as a manual but efficient alternative. Theoretically, this underscores the potential for curve fitting to serve as a standalone or supplementary solution in LED correction systems.

The theoretical implications extend to the concept of chromaticity accuracy and its dependence on measurement resolution. This study demonstrates that different measurement resolutions, represented by curve-fitting approaches, impact the accuracy of chromaticity correction. Higher resolutions, such as very high resolution ($n = 125$), provide the highest chromaticity accuracy but may involve significant time and effort. Conversely, the high-resolution approach ($n = 23$) offers a favorable balance between accuracy and practicality. This finding highlights the importance of choosing the right measurement resolution based on specific application requirements.

In real-world applications, such as projection mapping for visual enhancement, accurate and dynamic control of LED light sources is crucial for creating immersive and visually stunning experiences [35,36]. The findings of this study offer practical insights for optimizing chromaticity accuracy in projection mapping, enabling more lifelike and precise visuals. For visual enhancement applications, such as in entertainment and architectural lighting (or architainment), the ability to achieve accurate color reproduction is a key factor in enhancing user experiences.

In healthcare settings, particularly for cyanosis detection [37–39], accurate LED light sources are imperative for reliable diagnostic outcomes. The results of this research imply that, by choosing an appropriate curve-fitting resolution, healthcare facilities can enhance the accuracy of cyanosis detection systems without resorting to complex and costly control methods. This can lead to more effective and efficient diagnostic procedures, potentially saving lives.

Art conservation and restoration demand precise lighting conditions to protect and preserve valuable artworks [7,8,40–42]. Curve fitting's ability to correct nonlinear dimming responses in LED light sources can be invaluable for art conservation efforts. Achieving optimal chromaticity accuracy through curve fitting ensures that artwork is displayed and conserved under conditions that maintain its integrity and visual appeal as well as reduce energy consumption [43,44].

Circadian entrainment, which plays a crucial role in regulating human biological rhythms, relies on the precise control of lighting conditions to mimic natural daylight [45–49]. The research findings suggest that curve fitting can contribute to accurate lighting systems for circadian entrainment, potentially benefiting applications in healthcare facilities, workplaces, and residential settings where maintaining a healthy daily rhythm is essential.

Therefore, this study demonstrates the practicality of using curve fitting as a means of correcting the nonlinear dimming response of LED light sources. The theoretical implications highlight the trade-offs between complex and costly correction methods and the simplicity and cost-effectiveness of curve fitting. Furthermore, the application implications underscore the real-world significance of achieving optimal chromaticity accuracy in various domains, including projection mapping, healthcare, art conservation, and circadian entrainment.

## 5. Conclusions

SSL devices may exhibit a nonlinear response when dimmed, but the nonlinear response can be corrected by using a curve-fitting technique to increase the accuracy of light source optimization. In this study, the RGB channels of an LED projector were measured individually on each dimming step from 0 to 255. Since the computational time and effort to



first perform curve fitting and then optimize the light source spectrum can be tremendous, spectral data were analyzed with four different approaches, with a number of measurements ranging from low to very high resolution. The spectra estimated using four different curve-fitting approaches were compared with measured (actual) spectral output using the CIE 1976 ($u'v'$) chromaticity coordinates and RMSE. In addition, the theoretical (linear) output was also compared with the actual measured spectral output. It is important to report both perceptual metrics, such as chromaticity coordinates, and radiometric measures, such as RMSE, for a detailed comparison of the light source spectral output.

Curve fitting with very high resolution ($n = 125$, alternate dimming steps) decreased the chromaticity shifts between measured (actual) SPDs and corrected SPDs most significantly. Although the theoretical (linear) and corrected SPDs were statistically significantly different from measured (actual) data, the effect size calculations pointed out that high-resolution curve fitting ($n = 23$) performed equally well as very high-resolution approaches.

The novelty of the proposed approach is its independence from LED spectral composition, dimming, or manufacturer types. The method can be applied to various other SSL devices. Curve fitting can also serve as either a standalone solution or a supplementary measure in a variety of applications, including projection mapping, healthcare applications, art conservation, visual enhancement, and circadian entrainment. By adopting the appropriate curve-fitting resolution, users can achieve optimal chromaticity accuracy, translating to superior spectral output precision in these domains.

Nevertheless, this study does come with certain limitations. This study uses the RGB channels of an LED projector as proof of concept, leaving room for future exploration of curve fitting in different SSL device configurations. Additionally, the absence of considerations for warm-up and aging effects represents a limitation that future studies could address to enhance the accuracy of nonlinear dimming correction, particularly in prolonged use scenarios.

Future research has several intriguing avenues to explore. Incorporating the impact of warm-up and aging effects into curve-fitting calculations is a promising direction, as it can lead to more stable spectral output over time. The development of adaptive curve-fitting algorithms that can dynamically adjust the resolution based on real-time conditions, such as temperature variations or device aging, can further optimize SSL device performance. Integrating curve-fitting techniques into existing control systems, including open-loop and closed-loop methods, holds the potential for a seamless and effective approach to addressing nonlinearity while considering broader system dynamics. Researchers interested in this domain can utilize the curve-fitting method by using NIST-calibrated spectroradiometers in a controlled room and measure the radiometric output of multi-color LEDs. Finally, practical implementation considerations, including calibration, scalability, and compatibility with various hardware and control platforms, warrant further investigation to bring these curve-fitting solutions into real-world applications.

**Author Contributions:** Conceptualization, D.D.; methodology, D.D.; software, R.K.; formal analysis, R.K.; investigation, R.K.; resources, D.D.; data curation, R.K.; writing—original draft preparation, R.K.; writing—review and editing, D.D.; visualization, R.K.; supervision, D.D.; project administration, D.D. All authors have read and agreed to the published version of the manuscript.

**Funding:** This research received no external funding.

**Informed Consent Statement:** Not applicable.

**Data Availability Statement:** The data presented in this study are available on request from the corresponding author. The data are not publicly available due to storage issues.

**Conflicts of Interest:** The authors declare no conflict of interest.

## Nomenclature

| | |
|---|---|
| LED | Light-emitting diode |
| RGB | Red, green, blue |
| pcLED | Phosphor-coated LED |
| OLED | Organic LED |
| CCT | Correlated color temperature |
| PWM | Pulse-width modulation |
| PID | Proportional–integral–derivative |
| OL | Open loop |
| TFF | Temperature feed-forward |
| FFB | Flux feedback |
| CCFB | Color coordinates feedback |
| SNR | Signal-to-noise ratio |
| CSK | Color-shift keying |
| FOV | Field of view |
| Curve-fitting | The process of constructing a curve, or mathematical function, that has the best fit to a series of data points, possibly subject to constraints. |
| MATLAB | A proprietary multi-paradigm programming language and numeric computing environment. |
| CIE 1976 $(u'v')$ | A two-dimensional chromaticity diagram that allows light source chromaticity differences to be computed as a Euclidean distance. |
| MacAdam ellipses | Graphical representations of perceptible chromaticity differences plotted on a chromaticity diagram based on visual experiments conducted by David L. MacAdam [50]. |

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
