# Peer review of "Curve-Fitting Correction Method for the Nonlinear Dimming Response of Tunable SSL Devices"

_technologies, doi:10.3390/technologies11060162_

Round 1
Reviewer 1 Report
Comments and Suggestions for Authors
The research is very interesting and deals with a significant issue, the proper dimming of LEDs. The aim of the research fits to the scope of the journal. Some minor comments
There are some significant works related to the role of dimming in lighting (light output vs power consumption) with laboratory experiments that could enchance the referenses
In the conclusions the authors should make clear the novelty of their work along some basic steps for future research if other teams want to replicate the measurements
Author Response
Thank you very much for taking the time to review this manuscript. Please find the detailed responses below and the corresponding revisions/corrections highlighted/in track changes in the re-submitted files.
The research is very interesting and deals with a significant issue, the proper dimming of LEDs. The aim of the research fits to the scope of the journal. Some minor comments
Thank you.
There are some significant works related to the role of dimming in lighting (light output vs power consumption) with laboratory experiments that could enchance the referenses
We added references to enhance the literature review.
In the conclusions the authors should make clear the novelty of their work along some basic steps for future research if other teams want to replicate the measurements
We highlighted the novelty of this proposed method and indicated some potential research areas.
Reviewer 2 Report
Comments and Suggestions for Authors
The authors show that for a very specific – but not specified – LED projector in a specific measurement setup the proposed fitting procedure allows to optimize the dimming response – if I understood correctly - with respect to power and spectrum.
Being no expert in lightning nor in statistics, I would guess that the measurements and statistics have been done correctly – although one would need the data set for verification.
My main concern is that these findings are of very limited importance since they can readily only be applied to the specific projector used, which is not even specified. This point is finally addressed in the ‘Conclusions’: “… this study does come with certain limitations….”
To be of more relevance, this method should have been applied to at least two or three different light engines with different power control (via duty cycle or via dc current), heat management, etc., to demonstrate its universal usefulness.
At least, the authors should give detailed information on the light source used. For instance, the method of power control (current and/or pulse code modulation) and the heat sinking (separate for each color? Temperature stabilized?) make a big difference how the SPD changes spectrally, power wise and dynamically. Is the input to the LEDs linear in the dimming steps, or is the S-shape an electronic feature to enhance the contrast of the image? So, perhaps the “ideal (linear) fitting response” (Fig. 2, not shown) was actually not the ideal response for the manufacturer of this projector?
To get an idea of what is happening, I would suggest to show in Fig. ½ total/maximum-spectral power response (including fitting) of all three color-channels. Also, it would be helpful to see the spectra for low and high power in comparison.
Did I understand correctly, the fitting and the subsequent control was only done to the power per color channel (either the total power or the maximum spectral power), and the spectral shift of the emission was not taken into account? I. e., the “resolution” varied in the curve fitting applies to the power setting (0 … 255) only?
I assume, all the measurements were done stationary, i. e., in thermal equilibrium? What is the thermal response time if, e. g., the drive power is changed? Is there any crosstalk between the channels?
I wonder what would be the typical setup and procedure, which would be optimally suited for this type of spectral power control:
a) Measuring and fitting the correction parameters once and then applying these (stored) parameters in daily operation? However, in this case the advantage of having a fast fitting-process seems to be not really relevant.
b) Measuring and fitting during operation? In this case I wonder how one could go through the whole power curve to extract the fit parameters. What that be fast enough to correct also for temperature changes? And it would be less complex than “the well-known methods” mentioned in the ‘Discussion’.
Is it a good idea to use a fit which does not hit the origin (0/0) (see Fig. 1 & 2)? This may minimize the overall absolute RMS figure and work well for large output, but the relative error for small output reaches easily +/- 100 % and may ruin the color balance completely unless the error for each channel is similar.
In Tab. 1, is there any difference between the “Total power” and the “Average power” besides the scaling by the number of summed/averaged data points? This would explain why the corresponding R² numbers are identical.
In short: Since I assume it will not be possible to add one or two more (different) light sources to the analysis, I would strongly recommend to at least add detailed information on the light source used in the paper. In this way, the reader has a chance to guess to what type of light sources these findings can be applied to.
Reviewer 3 Report
Comments and Suggestions for Authors
The Reviewed article presents some interesting investigation results, but it needs revision before publication.
In the revised version of this article the Authors should take into account the following remarks:
1. Section Nomenclature including explanation of all the abbreviations should be added.
2. The aim of the article should be clearly stated in Introduction.
3. In the literature review some papers by P. Ptak and M. Janicki should be taken into account.
4. The used investigation method should be described in detail with adequate schematic diagram of the used set-up.
5. The presented results of investigations should be compared with other method describing in the literature.
6. In Conclusions the value of the performed investigations results should be clearly stated.
Author Response
Thank you very much for taking the time to review this manuscript. Please find the detailed responses below and the corresponding revisions/corrections highlighted/in track changes in the re-submitted files.
The Reviewed article presents some interesting investigation results, but it needs revision before publication.
In the revised version of this article the Authors should take into account the following remarks:
- Section Nomenclature including explanation of all the abbreviations should be added.
We added a nomenclature section.
- The aim of the article should be clearly stated in Introduction..
We added aims of the paper.
- In the literature review some papers by P. Ptak and M. Janicki should be taken into account.
We added reference to the work done by Ptak and Janicki.
- The used investigation method should be described in detail with adequate schematic diagram of the used set-up.
We added an image showing the set-up.
- The presented results of investigations should be compared with other method describing in the literature.
Unfortunately, previous research has not evaluated the dimming accuracy in Delta u’v’ and RMSE metrics (which was a limitation of past studies). Therefore, we couldn’t provide a comparison with previous research studies.
- In Conclusions the value of the performed investigations results should be clearly stated.
Done.

Round 2
Reviewer 3 Report
Comments and Suggestions for Authors
The Authors properly addressed my remarks. The article can be accepted. In the reference list the name of one author of the ref. [28] is neglected. It should be corrected before publication.